# A Local Graph Limits Perspective on Sampling-Based GNNs

## Abstract

We propose a theoretical framework for training Graph Neural Networks (GNNs) on *large* input graphs via training on *small*, *fixed-size* sampled subgraphs. This framework is applicable to a wide range of models, including popular sampling-based GNNs, such as GraphSAGE and FastGCN. Leveraging the theory of graph local limits, we prove that, under mild assumptions, parameters learned from training sampling-based GNNs on small samples of a large input graph are within an $\epsilon$-neighborhood of the outcome of training the same architecture on the *whole graph*. We derive bounds on the number of samples, the size of the graph, and the training steps required as a function of $\epsilon$. Our results give a novel theoretical understanding for using sampling in training GNNs. They also suggest that by training GNNs on small samples of the input graph, practitioners can identify and select the best models, hyperparameters, and sampling algorithms more efficiently. We empirically illustrate our results on a node classification task on large citation graphs, observing that sampling-based GNNs trained on local subgraphs $12\times$ smaller than the original graph achieve comparable performance to those trained on the input graph.

## 1 Introduction

As the size and complexity of graph data continue to increase, there is a growing need to find ways to scale Graph Neural Networks (GNNs). Yet, scaling GNNs to larger graphs faces two key obstacles: training inefficiencies due to repeated gradient calculations at every node; and large memory requirements for storing not only the graph but also the node embeddings. To overcome these challenges, a variety of efficient GNN training algorithms have been introduced which leverage a wide array of sampling techniques. Examples include GraphSAGE (Hamilton et al., 2017), FastGCN (Chen et al., 2018), and shaDoW-GNN (Zeng et al., 2021), among others (see (Ma et al., 2022a) for a thorough review).

Despite the success of sampling-based GNNs in practice, there are still no formal models that explain why they can reduce the computation and memory requirements of GNNs without significantly compromising their performance[1]. We take a step in this direction by proposing a theoretical framework that abstracts away the details of specific architectures and sampling algorithms and defines a general class of sampling-based GNNs (see Algorithm 1). In this framework, sampling is employed in two steps. First, nodes are sampled to estimate the gradient and update the GNN weights for the next iteration (*node sampling*). Second, a sampling method is used to prune the computational graph when computing the node embeddings via neighborhood aggregations (*computational graph sampling*).

As a second contribution, we propose to use this general framework to study a training procedure in which, instead of training the GNN on the whole graph, we train it on a collection of small subgraphs sampled from the input graph. Intuitively, this approximates both node sampling—as gradients are only computed for nodes in the subgraph—and computational graph sampling—as the computational subgraphs are not necessarily induced subgraphs[2]. We prove the validity of this approximation

---

[1] We use the term 'sampling-based GNN' broadly to refer to any GNN architecture that utilizes node sampling and/or computational graph sampling.

[2] Some papers in the early GNN literature introduced both a novel architecture and a sampling-based training algorithm. This has caused confusion as it is not uncommon for the architecture and the algorithm to be called

theoretically, by showing that training such sampling-based GNNs on small sampled subgraphs yields a similar outcome to training these GNNs directly on the large target graph (Theorem 5.4). The proof relies on the theory of graph limits (Aldous & Steele, 2004; Benjamini & Schramm, 2001), which provides a way to understand the behavior of a family of graphs with similar local structures. We view the large input graph as the infinite 'limit' of sampled graphs with similar local structures (i.e., similar motifs such as triangles and $k$-cycles). Then, we show that the training behavior on these smaller subgraphs converges to the training behavior on the limit graph.

In practice, our results apply to a variety of architectures, including GCN (Kipf & Welling, 2017), GraphSAGE, and GIN with mean readout (Xu et al., 2019); and a variety of gradient and computational graph sampling schemes, including the neighborhood sampling scheme from GraphSAGE, FastGCN, and shaDoWGNN (Zeng et al., 2021) (Corollary 5.6 and Theorem 5.7). Therefore, our results significantly extend previous results on GNN convergence, which focused almost exclusively on convolutional GNNs (Ruiz et al., 2020; Keriven et al., 2020; Roddenberry et al., 2022). Our theoretical findings also provide a practical guideline for training GNNs on large graphs more efficiently. They suggest that practitioners can compare different sampling-based GNNs and their choice of hyperparameters on small samples of the input graph, knowing that these small samples are good approximations for training on the entire input graph as long as their underlying models and architectures fit within our framework.

We demonstrate the validity of our results empirically on two node classification tasks: on large citation graphs with up to 20,000 nodes (Section 6); an on a *very* large citation network from the ogbn-mag graph, with 200,000 nodes. We observe that various sampling-based GNNs trained on local subgraphs achieve comparable performance to those trained on the large input graph, even when the subgraphs are up to $40\times$ smaller than the original graph.

**Summary.** Our main contributions are:

- A unifying framework for sampling-based GNNs (Algorithm 1) incorporating gradient and computational graph sampling.
- A simplified training procedure to approximate sampling-based GNNs, wherein GNNs are trained on small subgraphs sampled from the original input graph.
- Proving, using the theory of graph limits, that training GNNs on small samples of the large input graph yields learned parameters within an $\epsilon$-neighborhood of training the GNN on the large target graph (Theorems 5.4 and 5.7). This can be interpreted to mean that sampling-based GNNs produce similar outcomes as training on the original input graph.
- Extending GNN convergence results to many well-known architectures such as GCN, GraphSAGE and GIN, and many node and computational graph sampling schemes such as neighborhood sampling (GraphSAGE), FastGCN, and shaDoW-GCN (Corollary 5.6 and Theorem 5.7).
- Empirical results on a node classification task on PubMed and ogbn-mag, in which we observe that sampling-based GNNs trained on local subgraphs achieve comparable performance to those trained on the large target graph (Section 6).

## 2 RELATED WORK

**Sampling-based GNNs.** Stochastic node sampling for GNN training is inspired by minibatch stochastic gradient descent and was first proposed by Hamilton et al. (2017). Hamilton et al. (2017) also introduced GraphSAGE, which can be trained via random neighborhood sampling, a type of computational graph sampling detailed in Appendix F. Aiming to further decrease complexity, FastGCN uses importance sampling to sample the computational graph (Chen et al., 2018), shaDoW-GCN first samples a collection of subgraphs of depth $K$—instead of sampling during training—and then trains a conventional GNN on them (Zeng et al., 2021). Our framework also applies to more recent advancements on sampling-based GNNs such as GNNAutoScale (Fey et al., 2021), GraphFM (Yu et al., 2022), LMC (Shi et al., 2022), and IBMB (Gasteiger et al., 2022). These architectures and sampling mechanisms are discussed in further detail in Appendix F. See also (Liu et al., 2021) for a complete survey on sampling-based GNNs.

---

by the same name, e.g., GraphSAGE can refer to both the architecture and the computational graph sampling scheme proposed by Hamilton et al. (2017). In general, when using these names, we will be referring to the sampling algorithm. We will explicitly specify it when referring to the architecture.

A different related line of work is neural network pruning at the training stage, which is a technique for reducing the computational complexity and memory requirements of deep neural networks (Zhu & Gupta, 2017; Gale et al., 2019; Strubell et al., 2019). In this line of work, they make training more effective and efficient by sampling the connections between neurons of a neural network (while in our work we study sample data that has a network structure). Recent works on the lottery ticket hypothesis (Frankle & Carbin, 2018; Frankle et al., 2020) have shed light on the possibility of sampling subnetworks during the early phases of training that achieve comparable accuracy to the full network while significantly reducing its complexity.

Tangential to our work is (Yehudai et al., 2021) explores distribution shifts in graph structures, emphasizing potential generalization issues. In contrast, we focus on unifying frameworks for training sampling-based GNNs and aim to prove the convergence of training across graph families with consistent local structures.

**Benjamini-Schramm convergence.** The theory of graph limits introduced by Benjamini & Schramm (2001); Aldous & Steele (2004) has been used for studying random network models. Almost all sparse random graph models, including Erdös-Rényi graphs (van der Hofstad, 2021, Theorem 2.17), configuration models (Dembo & Montanari, 2010) (van der Hofstad, 2021, Theorem 4.5), preferential attachment models (Berger et al., 2014), geometric random graphs (Bollobás et al., 2007), and motif-based models (Alimohammadi et al., 2022) are known to have graph limits.

## 3 SAMPLING-BASED GNNS

### 3.1 GNNS: PRELIMINARIES

Let $G_n = (V(G_n), E(G_n))$ be a graph where $V(G_n)$, $|V(G_n)| = n$, is the set of nodes, $E(G_n) \subseteq V(G_n) \times V(G_n)$ is the set of edges, and $\mathbf{A} \in \mathbb{R}^{n \times n}$ is the adjacency matrix. Let $\mathbf{X} \in \mathbb{R}^{n \times F}$ be the matrix of input features, where $\mathbf{x}_v$ is the $F$-dimensional feature vector of node $v$.

In its most general form, a GNN consists of $L$ layers, each of which composes an *aggregation* of the information in each node's neighborhood, and a *combination* of the aggregate with the node's own information. Explicitly, for each layer $\ell + 1$ and node $v$ we can write the following propagation rule (Xu et al., 2019)

$$
\begin{aligned}
\mathbf{h}_{N(v)}^{(\ell+1)} &= \texttt{AGGREGATE}_\ell \left( \{ \mathbf{h}_u^{(\ell)}, u \in N(v) \} \right) \\
\mathbf{h}_v^{(\ell+1)} &= \texttt{COMBINE}_\ell \left( \mathbf{h}_v^{(\ell)}, \mathbf{h}_{N(v)}^{(\ell+1)} \right) \qquad \text{and} \qquad \mathbf{h}_v^{(0)} = \mathbf{x}_v,
\end{aligned}
\tag{1}
$$

where $N(v)$ is the neighborhood of node $v$ and $\mathbf{H}^{(\ell)} \in \mathbb{R}^{n \times F_\ell}$ is the embedding produced by layer $\ell$.

In so-called node-level or inductive learning tasks, the GNN output is $\mathbf{Z} = \mathbf{H}^{(L)}$. In graph-level or transductive learning tasks, the GNN has a final layer called the *readout* layer, which aggregates the node embeddings $\mathbf{h}_v^{(L)}$ into a single graph embedding $\mathbf{h}_G \in \mathbb{R}^{F'}$ as follows

$$
\mathbf{h}_G = \texttt{READOUT}(\mathbf{h}_v^{(L)}, v \in V(G))
\tag{2}
$$

and where $F'$ is the embedding dimension. The output of the GNN is then $\mathbf{Z} = \mathbf{h}_G$. The READOUT function can be a fully connected layer over the graph nodes, a sequence of pooling layers (Ying et al., 2018; Zhang et al., 2018), or a simple aggregation operation such as the maximum, the sum, or the average (Xu et al., 2019). We focus on architectures where the READOUT layer (if present) is a simple aggregation operation, such as in (Xu et al., 2019) and (Dwivedi et al., 2020). This is a common assumption in the literature, since such aggregations are invariant to node relabelings and prevent the number of learnable parameters from depending on the size of the graph.

For examples of how two popular GNN architectures, GCN and GraphSAGE, can be written in the form of (1)–(2), refer to Appendix B. We will experiment with these architectures in Section 6.

**Training.** We consider both unsupervised and supervised learning tasks. In the unsupervised case, the goal is to minimize a loss $\mathcal{L}$ over the dataset $\mathcal{T} = \{\mathbf{X}_m\}_m$. In the supervised case, each data point is additionally associated with a label $\mathbf{Y}_m$, which factors in the computation of the loss. In

either case, the goal is to solve the following optimization problem

$$\min_{\mathbf{W}^{(\ell)}, 1 \le \ell \le L} \frac{1}{|\mathcal{T}|} \sum_{m=1}^{|\mathcal{T}|} \mathcal{L}(\mathbf{Z}_m) \tag{3}$$

where $\mathbf{Z}_m$ are the outputs of the GNN corresponding to the inputs $\mathbf{X}_m$.

This problem is solved using a gradient descent approach with updates

$$\mathbf{W}_{t+1}^{(\ell)} = \mathbf{W}_t^{(\ell)} - \frac{\eta}{|\mathcal{T}|} \sum_{m=1}^{|\mathcal{T}|} \nabla_{\mathbf{W}} \mathcal{L}(\mathbf{Z}_m) \tag{4}$$

where $\eta$ is the step size or learning rate. The process stops when the gradient becomes smaller than some predetermined small constant $\sum_m |\nabla_{\mathbf{W}} \mathcal{L}| \le |\mathcal{T}| \epsilon$.

Computing the loss (3) and the gradients (4) can be cumbersome when the graph $G$ is large, but as we discuss in the next section, both can be estimated using sampling techniques.

## 3.2 SAMPLING-BASED GNNS: AN GENERAL FRAMEWORK

We consider a variety of sampling techniques under a general unified framework. In particular, we offer a formalization for sampling-based GNNs that utilize sampling in one or both of two ways: node sampling and computational graph sampling.

**Node sampling.** Due to the difficulty of computing gradient descent steps for every node on a large graph, a common technique to accelerate GNN training is to perform stochastic gradient descent (SGD) over minibatches of graph nodes. SGD in its conventional form samples a minibatch of nodes $V_B \subset V(G)$ and then uses the gradient on these nodes to estimate (4). I.e., it uses $\frac{1}{|V_B|} \sum_{v \in V_B} \nabla_{\mathbf{W}} \mathcal{L}(\mathbf{z}_v)$ as an estimator for $\nabla_{\mathbf{W}} \mathcal{L}(\mathbf{Z})$.

Other variants of SGD employ importance sampling to estimate the gradient. Let $\nu_g : G \to \mathbb{R}^+$ be a weight function influencing the sampling probability as $\frac{\nu_g(G,v)}{|V(G)|}$. Then,

$$\nabla_{\mathbf{W}} \tilde{\mathcal{L}}_{\nu_g}(\mathbf{Z}) = \frac{1}{|V_B|} \sum_{v \in V_B} \frac{1}{\nu_g(v)} \nabla_{\mathbf{W}} \mathcal{L}(\mathbf{z}_v) \tag{5}$$

gives an estimator for (4). Note that if $\nu_g(v) = 1$ for all $v \in V(G)$, we recover conventional SGD.

**Computational graph sampling.** In conventional GNN training, i.e., without sampling, the size of the computational graph grows exponentially with the number of layers. Hence, many architectures prune the computational graph by sampling which neighbor-to-neighbor connections to keep (those that aren't sampled are discarded). Specifically, a computational graph sampler $\nu_C$ taking a graph $G$ and a node $v \in V(G)$ as inputs outputs a sampled computational graph denoted $\nu_C(G, v)$ (or simply $\nu_C(v)$ if $G$ is clear from context). This sampled computational graph, $\nu_C(G, v)$, then replaces the full computational graph in the forward propagation step (1).

Having established these sampling processes, we now proceed to describe our proposed unified algorithmic framework which uses a node sampler to compute gradients and a computational graph sampler to compute the forward pass.

**A unified algorithmic framework for sampling-based GNNs.** A sampling-based GNN takes as inputs a graph, a gradient sampler $\nu_g$, and a computational graph sampler $\nu_C$. It then samples a minibatch of nodes $V_B$ from the graph using $\nu_g$; uses $\nu_C$ to sample a computational graph $G_V$; performs forward propagation on $G_V$; and outputs the embeddings $\mathbf{Z}$ of the sampled nodes $V_B$. This is shown in Algorithm 1, where we write the outer for loop for explanation purposes only. In practice, the steps in this outer loop are executed in parallel.

This framework encompasses many well-known GNN architectures. For instance, in GraphSAGE (Hamilton et al., 2017), the node sampler $\nu_g$ assigns a uniform probability to all nodes, and the computational graph sampler $\nu_C$ draws a fixed number of neighbors for each node. A second example is FastGCN, which prunes the computational graph by choosing neighbors proportionally to the normalized adjacency matrix entries. We provide a more detailed discussion on how various architectures, including GraphSAGE, FastGCN and shaDoWGNN, fit into this algorithm in Appendix F.

---

**Alg. 1:** Unified Framework for Sampling-Based GNN

---

**Function** `SamplingBasedGNN`
**Input:** graph $G_t$; gradient sampler $\nu_g$; comp. graph sampler $\nu_C$

Sample minibatch of $|V_B|$ nodes $v \sim \nu_g(v)$
**for** $v \in V_B$ **do**
    Sample computational graph $G_v \sim \nu_C(v)$
    **for** $\ell = 0$ *to* $L - 1$ **do**
        $\mathbf{h}_{N(v)}^{(\ell+1)} = \text{AGGR}_\ell(\{\mathbf{h}_u^{(\ell)}, (u,v) \in E(G_v)\}$
        $\mathbf{h}_v^{(\ell+1)} = \text{COMBINE}_\ell\left(\mathbf{h}_{N(v)}^{(\ell+1)}, \mathbf{h}_v^{(\ell+1)}\right)$
    $\mathbf{Z} = \mathbf{H}^{(L)}$ or $\text{READOUT}(\mathbf{h}_v^{(L)}, v \in V(G))$
**return Z**

---

**Alg. 2:** Training by Sampling Local Subgraphs

---

**Input:** sample size $N_\epsilon$; subgraph sampler $\mu_S$

**while** $|\nabla \mathcal{L}| > \epsilon$ **do**
    Draw $N_\epsilon$-node graph $G_t \sim \mu_S$
    $\mathbf{Z} = \texttt{SamplingBasedGNN}(G_t)$
    $\nabla_{\mathbf{w}_t}\hat{\mathcal{L}}_{\nu_g}(\mathbf{Z}) = \frac{1}{|V_B|}\sum_{v \in V_B}\frac{1}{\nu_g(v)}\nabla\mathcal{L}(\mathbf{z}_v)$
    $\mathbf{W}_{t+1} = \mathbf{W}_t - \eta\nabla_{\mathbf{w}_t}\nabla\tilde{\mathcal{L}}_{\nu_g}$
    $t \leftarrow t + 1$

---

**Training by sampling local subgraphs.** Typically, sampling-based GNNs are trained for a fixed number of rounds where each round consists of running Algorithm 1 followed by a backward pass (i.e., weight updates via some variant of gradient descent; see full description in Algorithm 3 in the appendices). We study a slight modification in which, instead of training the GNN on the full graph, we train it on a collection of smaller subgraphs. This is achieved by employing a subgraph sampler $\mu_S$ which acts as an oracle: it subsamples graphs that are then passed to the sampling-based GNN.

This change in the training procedure allows analyzing the training convergence of sampling-based GNNs without significantly changing the original algorithm. Indeed, we will show that, above a certain lower bound on the size of the subgraphs produced by $\mu_S$, Algorithm 2 reaches the neighborhood of a local minimum in a finite number of training steps.

## 4    Graph Limits and Limit GNNs

We introduced a training procedure for sampling-based GNNs which consists of training them on a collection of local subgraphs that are sampled at regular intervals during training. The purpose of this change in training procedure is to approximate the effect of node and computational graph sampling. However, some questions remain unanswered. Specifically, does this approach yield results similar to training on the entire input graph? Do the sampled subgraphs contain enough information about the graph, such that training is not significantly affected? If we are able to give positive answers to these questions, we can analyze sampling-based GNNs on smaller graphs, which in turn allows for better interpretability and can facilitate the design and tuning of GNNs by practitioners. To answer these questions, we turn to the theory of graph limits. We first introduce graph limits and the associated notion of convergence, before defining limit GNNs.

### 4.1    Graph Limit Theory

At a high level, a sequence of graphs $\{G_n\}_{n \in \mathbb{N}}$ is said to converge locally if the empirical distribution of the local neighborhood of a uniform random node converges. The original definition of local convergence by Benjamini & Schramm (2001) applies to graphs with no node or edge features. Here, we consider graphs with attributes where each node and edge is associated with an input feature and (in the case of supervised learning) a target feature. This is reminiscent of 'marked graph convergence' (Benjamini et al., 2015) (van der Hofstad, 2021, Ch. 2).

To formalize the definition of local convergence, let $(G, o)$ denote a rooted graph with attributes, which is a graph $G$ with node/edge attributes to which we assign a root node $o$. Let $\mathcal{G}_*$ be the set of all possible rooted graphs with attributes. A limit graph is defined as a measure over the space $\mathcal{G}_*$ with respect to a local metric $d_{loc}$. For a pair of rooted graphs $(G_1, o_1)$ and $(G_2, o_2)$, the distance $d_{loc}$ is given by

$$d_{loc}((G_1, o_1), (G_2, o_2)) = \frac{1}{1 + \inf_k\{k : B_k(G_1, o_1) \not\simeq B_k(G_2, o_2)\}},$$

where $B_k(G, v)$ is the $k$-hop neighborhood of node $v$ in graph $G$, and $\simeq$ represents the graph isomorphism. Since the limit graph is rooted, we need to make finite graphs in the sequence $\{G_n\}_{n \in \mathbb{N}}$ rooted as well by choosing a uniform random root denoted $\mathcal{P}_n = \frac{1}{n}\sum o_n \in V(G_n)\delta(G_n, o_n)$.

**Definition 4.1** (Local Convergence with Attributes). Let $\mu$ be a measure on the space $\mathcal{G}_*$. Then, a sequence of graphs $\{G_n\}_{n \in \mathbb{N}}$ is said to converge locally in probability to a graph $\mu \sim \mathcal{G}_*$ if, for any $k > 0$ and any finite graph $Q$ with nodes at most $k$ hops from the root,

$$\mathbb{P}_{v \sim \mathcal{P}_n}[Q \sim B_k(G_n, v)] \xrightarrow{\mathbb{P}} \mathbb{P}_\mu(Q \sim B_k(G, o)).$$

Equivalently, for any bounded and continuous (with respect to metric $d_{loc}$) function $f : \mathcal{G}_* \to \mathbb{R}$,

$$\mathbb{E}_{v \sim \mathcal{P}_n}[f(G_n, v)|G_n] \xrightarrow{\mathbb{P}} \mathbb{E}_\mu[f(G, o)]. \tag{6}$$

The above definition points out the equivalence between the convergence of local neighborhoods around random nodes and the convergence of bounded and continuous functions known as local functions (for the proof of equivalence, see (van der Hofstad, 2021, Ch. 2)). Intuitively, a local function applied to finite rooted samples of a graph limit can be shown to converge to the function applied directly to the infinite graph limit. Building upon this idea, we define *almost local functions* as follows.

**Definition 4.2** (Almost Local Functions). A function $f : \mathcal{G}_*^N \to \mathbb{R}^K$ is said to be *almost local* if, for a sequence of graphs $\{G_n\}_{n \in \mathbb{N}}$ converging to a limit graph $\mu \sim \mathcal{G}_*$, it converges to a limit function $\tilde{f} : \mathcal{G}_*^N \to \mathbb{R}^K$ as

$$\mathbb{E}_{v_1, v_2 \ldots v_N \sim \mathcal{P}_n}\left[ f\big((G_n, v_1), \ldots, (G_n, v_N)\big)|G_n \right] \xrightarrow{\mathbb{P}} \mathbb{E}_{(G^{(i)}, o_i) \sim \mu}\left[ \tilde{f}\big((G^{(1)}, o_1), \ldots (G^{(N)}, o_N)\big) \right]. \tag{7}$$

**Remark.** The definition above departs slightly from the conventional definition of local functions in the literature, which typically requires boundedness and continuity. Still, the introduction of almost local functions is necessary for the analysis of sampling-based GNNs. For instance, many typical sampling methods assign weights to nodes based on global parameters of the graph such as the average moments of the degree, which violates the local continuity condition. Similarly, loss functions based on negative sampling (e.g., (Hamilton et al., 2017)) are not local functions in the conventional sense as they depend on the embedding of vertices located far away from the root. In Appendix D, we prove that both normalized adjacency sampling and loss functions with negative sampling yield functions that are almost local.

## 4.2 GNN in the limit

We define limit GNNs, i.e., GNNs on infinite graph limits, by extending the aggregate-readout architecture in (1) to infinite graphs.

**Definition 4.3** (Limit GNNs). Consider a (possibly infinite) rooted graph $(G, o)$ drawn from $\mathcal{G}*$ following distribution $\mu$. Given a GNN with $L$ layers, consider an $L$-neighborhood of the root $o$, denoted $B_L(G, o)$. Let $\mathbf{Z}_o$ be the output of the $L$-layer GNN given graph $B_L(G, o)$ [cf. (1)]. Then, the output embedding of the limit GNN is $\mathbb{E}(G, o) \sim \mu[\mathbf{Z}_o]$.

In the next section, we use this limit GNN to analyze sampling-based GNNs fitting the description of Algorithm 2.

## 5 Convergence of Sampling-Based GNNs

Our main result demonstrates that a sampling-based GNN trained on a collection of subgraphs sampled from the large target graph converges to an $\epsilon$-neighborhood of optimal limit GNN, i.e., the GNN that would be obtained by training on the full graph. We give convergence results for transductive and inductive learning tasks, and show their application to commonly used GNN architectures.

Our results rely on two sets of assumptions. First, we require the loss function to be bounded and Lispchitz continuous. This assumption is not very stringent, and is commonly used in the literature.

**Assumption 5.1** (Loss Function). The loss function $\mathcal{L}$ is bounded. Further, the loss function $\mathcal{L}$ and its gradient $\nabla \mathcal{L}$ are Lipschitz in the learning coefficients $\mathbf{W}$ with Lipschitz constant $C$.

Second, we require the collection of subgraphs on which the GNN is trained to satisfy the conditions of local convergence given in Section 4. We further assume that the sampling methods in Algorithm 2 are almost local, and hence can be defined on the graph limit.

**Assumption 5.2** (Almost Local Loss, Sampler and Aggregators)**.** The computational graph sampling scheme $\nu_C$, the loss function $\mathcal{L}$, and COMBINE$_\ell$ and AGGREGATE$_\ell$ for $\ell \in [0, L]$ are almost local.

**Assumption 5.3** (Convergent Sequence of Graphs)**.** The sequence of graphs $\{G_n\}_{n \in \mathbb{N}}$ converges locally in probability to $(G, o) \sim \mu$, where $\mu$ is a probability measure on the space of rooted graphs $\mathcal{G}_*$.

**Theorem 5.4.** *Consider a sampling-based GNN with L layers [cf. Algorithm 2] and with uniform node sampler $\nu_g$ (SGD), and additionally satisfying Assumptions 5.1–5.2. Let the collection of subgraphs on which the GNN is trained (generated by sampler $\mu_S$) define a convergent graph sequence $\{G_n\}_{n \in \mathbb{N}}$ as in Assumption 5.3. Then, there exists a learning rate $\eta > 0$ such that:*

*1. For any $\epsilon > 0$, there exists $N_\epsilon > 0$ such that training Algorithm 2 on subgraph samples of size at least $N_\epsilon$ converges to the $\epsilon$-neighborhood of the optimal GNN on the limit G.*
*2. The expected number of training steps, ad therefore the expected number of subgraph samples needed for convergence, is $\tilde{O}(\frac{1}{\epsilon^2})$.*

Let $\mathbf{W}_t$ denote the GNN coefficients learned in iteration $t$ of Algorithm 2, i.e., the weights of the GNN trained on the collection of random subgraphs. Convergence to the $\epsilon$-neighborhood of the optimal limit GNN means that, after a finite number of training steps, the expected gradient of the loss in the limit graph, using these same coefficients $\mathbf{W}_t$, is bounded by $\epsilon$; or, explicitly, that $\mathbb{E}_{\mu, W_t}(|\nabla_{\mathbf{W}_t} \mathcal{L}(W_t, G)|) \leq \epsilon$. Here, note that the randomness arises from both the limit $(G, o) \sim \mu$ and the random initialization of the coefficients $\mathbf{W}_0$.

As a special case of this theorem, we focus on the situation in which the collection of subgraphs generated by $\mu_S$ is a collection of local subgraphs. By that we mean that $\mu_S$ first samples the infinite graph $(G, o) \sim \mu$ and then returns the breadth-first local neighborhood of size $N_\epsilon$ around the sampled root. If we think of the large target graph as the graph limit, the following result states that training on small subgraphs sampled via breadth-first search (BFS) is enough to get to the $\epsilon$-neighborhood of the optimal GNN on the large graph.

**Corollary 5.5.** *Given a sampling-based GNN satisfying the assumptions of Theorem 5.4, let the subgraph sampler $\mu_S = B_{N_\epsilon}(\mu(G, o)$ be a local BFS sampler. Then, the result of Theorem 5.4 holds.*

This result is general and applies to various GNN architectures, including but not limited to GCN (Kipf & Welling, 2017), GraphSAGE (Hamilton et al., 2017), FastGCN (Chen et al., 2018), and shaDoW-GNN (Zeng et al., 2021).

**Corollary 5.6.** *GCN, GraphSAGE, FastGCN, and shaDoW-GNN satisfy Assumption 5.2. Therefore, under Assumption 5.1, the results of Theorems 5.4 and Corollary 5.5 hold.*

This corollary has an important practical implication: it gives guarantees allowing practitioners to compare any of these GNNs by training them on small samples from the large target graph, which is much less costly than doing so on the large graph itself. In Appendix F, we provide detailed explanations for how each of these models fits into the unified framework of Algorithm 1 and satisfies the assumptions of our main result.

We conclude by showing that our result also applies to graph learning or transductive graph machine learning tasks. To do so, we need an additional assumption on the READOUT layer in (2), which must be almost local. This enables extending our results to even more architectures, such asGIN (Xu et al., 2019) with mean aggregation in the readout layer.

**Theorem 5.7.** *For a transductive sampling-based GNN satisfying the assumptions of Theorem 5.4 and with an almost-local READOUT layer, the result of Theorem 5.4 holds.*

## 6 EXPERIMENTS

We validate our results empirically through two sets of experiments: an ablation study of node and computational graph sampling on a large citation network (PubMed, ∼20k nodes); and a more realistic example on a very large citation network (sample from ogbn-mag, 200k nodes). Dataset

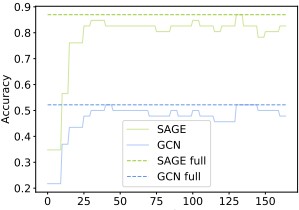 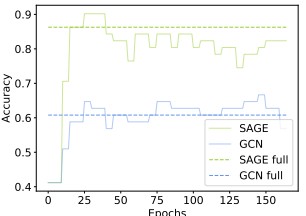 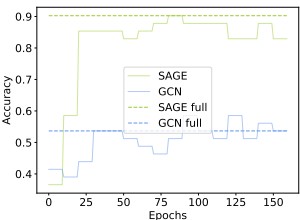

Figure 1: Node sampling results for PubMed with batch size 32. We consider three scenarios in terms of the small graph size $n$ and the graph sampling interval $\gamma$: $n = 1500$, $\gamma = 15$ epochs (left); $n = 2000$, $\gamma = 15$ epochs (center); $n = 1500$, $\gamma = 10$ epochs (right). Note that these graphs have size equal to approximately $10\%$ of the original graph size.

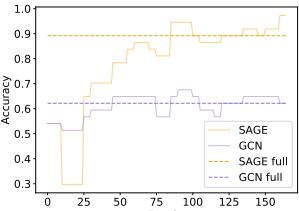 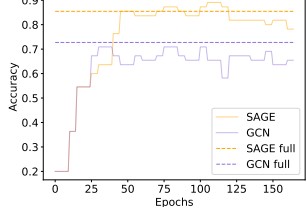 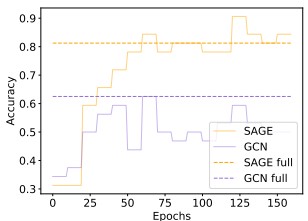

Figure 2: Computational graph sampling results for PubMed with $K_1 = K_2 = 32$ and batch size 32. We consider three scenarios in terms of the small graph size $n$ and the graph sampling interval $\gamma$: $n = 1500$, $\gamma = 15$ epochs (left); $n = 2000$, $\gamma = 15$ epochs (center); $n = 1500$, $\gamma = 10$ epochs (right). Note that these graphs have size equal to approximately $10\%$ of the original graph size.

details are provided in Appendix G. Common hyperparameters and training details are described in Appendix G, and details specific to each experiment are listed in the corresponding sections.

## 6.1 ABLATION STUDY ON A LARGE GRAPH

In first the experiment in this section, we consider node sampling independently, and then incorporate computational graph sampling in the second experiment. In each case, we compare GNNs trained on the full $N$-node graph with the same type of GNN, but trained on a collection of $n$-node subgraphs. These subgraphs are sampled via breadth-first search from random seeds sampled at random from the full graph at regular training intervals $\gamma$ (in epochs). In Figures 1–2, the dashed lines correspond to the best test accuracy of the model trained on the full $N$-node graph. The solid lines are the per epoch test accuracy of the models trained on the collection of stochastic $n$-node subgraphs.

**Node sampling.** We consider GNNs trained using the uniform random node sampling strategy (SGD) described in Section 3 and used, e.g., in (Hamilton et al., 2017). This sampling technique consists of partitioning the nodes into batches and, at each step, only considering the nodes in the current batch to compute the gradient updates. In Figure 1, we consider three scenarios in terms of the small graph size $n$ and the graph sampling interval $\gamma$: $n = 1500$, $\gamma = 15$ epochs (left); $n = 2000$, $\gamma = 15$ epochs (center); $n = 1500$, $\gamma = 10$ epochs (right). Note that these graphs have size equal to approximately $10\%$ of the original graph size. The GNNs are trained for $150$ epochs with learning rate $1e{-}4$ and batch size 32. In Figure 1, we observe that the GNNs trained on collections of random subgraphs achieve comparable performance to the GNNs trained on the full graph. Increasing the graph size $n$ leads some improvement for the GCN, but causes GraphSAGE to overfit. Increasing the sampling rate increases the variation in accuracy, but leads to a slight improvement in performance for both architectures.

**Computational graph sampling.** Next, we consider GNNs which, in addition to employing node sampling, use the computational graph sampling strategy proposed by Hamilton et al. (2017), called neighborhood sampling (see Appendix F). This technique consists of fixing parameters $K_\ell$ and, at each layer $\ell$, randomly sampling $K_\ell$ neighbors of each node from which to aggregate information.

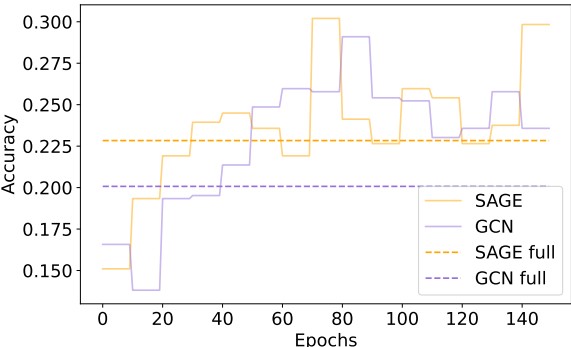

Figure 3: Computational graph sampling results for ogbn-mag with $K_1 = K_2 = 25$ and batch size 128. We consider one scenario in terms of the small graph size $n$ and the graph sampling interval $\gamma$: $n = 5000$, $\gamma = 10$ epochs.

We fix $K_1 = K_2 = 32$. The combinations of graph size $n$ and sampling interval $\gamma$ as well as the number of epochs and the learning rate are the same as in the previous experiment. The results are reported in Figure 2. We see that GNNs trained on collections of random subgraphs achieve comparable performance to GNNs trained on the full graph. Further, we observe some improvement in performance for GraphSAGE when the graph size is larger (center), and more variability in accuracy for both models when we increase the random subgraph sampling rate (right).

Additional ablation results for two other citation networks—Cora and CiteSeer—can be found in Appendix H. We also provide additional results for GNNs without node or computational graph sampling in Appendix I.

## 6.2 APPLICATION EXAMPLE ON A VERY LARGE GRAPH

Next, we consider a more realistic example on a very large graph: ogbn-mag. ogbn-mag is a heterogeneous citation network with 1,939,743 nodes representing authors, papers, institutions and fields of study (Hu et al., 2020). We focus exclusively on the paper-to-paper citation graph, which has 736,389 nodes. Due to memory limitations, we subsample it to 200,000 nodes. We consider a single experimental scenario using both node sampling with batch size 128 and computational graph sampling with $K_1 = K_2 = 25$. The learning rate was $1e - 2$.

The experiment results are reported in Figure 3. As before, the solid line corresponds to the GNN trained on a collection of randomly sampled subgraphs of size $n$, and the dashed line to the one trained on the full graph. We choose $n = 5000$ and resampling interval $\gamma = 10$ epochs for the former. Note that this choice of $n$ gives graphs with size equal to approximately 2.5% of the size of full graph.

## 7 CONCLUSION AND FUTURE STEPS

We have presented a novel theoretical framework for training GNNs on large graphs by leveraging the concept of local limits. Our algorithm guarantees convergence to an $\epsilon$-neighborhood of the learning GNN on the limit in $\mathcal{O}(1/\epsilon^2)$ training steps, which makes it a promising method for comparing different GNN architectures on large graphs efficiently.

Moving forward, a promising avenue for research is to explore the algorithm's robustness with adaptive sampling strategies, like those in ASGCN (Huang et al., 2018), or with schemes that globally encompass the graph, exemplified by ClusterGCN (Chiang et al., 2019). It would be interesting to see how the convergence behavior of the algorithm is affected when the limit itself changes over time. Additionally, it would be valuable to explore whether the weights learned by sampling-based GNNs in the limit can be coupled with the actual GNN. Although it may not be possible in general, under certain assumptions, it may be feasible to establish such a connection, opening up new avenues for further research in this field.

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

## A    MORE RELATED WORK

**Convergence and transferability of GNNs.** The convergence of GNNs on sequences of graphs, also called transferability, has been studied in a number of works. Ruiz et al. (2020) derive a non-asymptotic error bound for GNNs on dense graph sequences converging to graphons, and use it to show that GNNs can be trained on graphs of moderate size and transferred to larger graphs. These bounds are refined, and proved for a wider class of graphs, in (Ruiz et al., 2022a). Levie et al. (2021) prove convergence and transferability of GNNs on graphs sampled from general topological spaces, and Maskey et al. (2023) particularize this analysis to graphons. Keriven et al. (2020) study the convergence and transferability of GNNs on graphs sampled from a random graph model where the edge probability scales with the graph size, allowing for graphs that are moderately sparse. Roddenberry et al. (2022) define local distributions of the neighborhoods of a graph to prove a series of convergence results for graph convolutional filters, and in particular their transferability across graphs of bounded degree. Note that this is different than what we do in this paper: we focus on *sampling-based* GNNs, and prove convergence of their training on local subgraphs to GNNs trained on the large limit graph. More generally, random graph models have been a popular tool for understanding theoretical properties of GNNs, with several applications to topics including expressive power in community detection (Ruiz et al., 2022b), linear separability in semi-supervised classification (Baranwal et al., 2023), heterophily (Ma et al., 2022b).

**Pruning neural networks.**    Another line of research closely related to our work revolves around neural network pruning during the training stage. This approach aims to improve training efficiency by selectively sampling connections between neurons (Zhu & Gupta, 2017; Gale et al., 2019; Strubell et al., 2019). Notably, recent studies on the lottery ticket hypothesis (Frankle & Carbin, 2018; Frankle et al., 2020) have demonstrated that by sampling subnetworks during training, comparable accuracy to the full network can be achieved while significantly reducing its complexity. In contrast, our work takes a divergent trajectory as we shift our focus towards sampling data characterized by an intrinsic network structure, instead of manipulating the neural network connections.

## B    GCN AND GRAPHSAGE

For illustrative purposes, we focus on the GCN and GraphSAGE architectures. We also experiment with these architectures in Section 6.

**GCN.** The layerwise propagation rule of GCN is given by (Kipf & Welling, 2017)

$$\mathbf{H}^{(\ell+1)} = \sigma\left(\mathbf{D}^{-1/2}\mathbf{A}\mathbf{D}^{-1/2}\mathbf{H}^{(\ell)}\mathbf{W}^{(\ell+1)}\right) \qquad \text{and} \qquad \mathbf{H}^{(0)} = \mathbf{X}, \tag{8}$$

where $\mathbf{D} = \text{diag}(\mathbf{A1})$ is the degree matrix; $\mathbf{W}^{(\ell+1)} \in \mathbb{R}^{F_\ell \times F_{\ell+1}}$ are the learnable convolution weights at layer $\ell + 1$; and $\sigma$ is a pointwise nonlinearity such as the ReLU or the sigmoid. The adjacency matrix $\mathbf{A}$ is modified to include self-loops, hence the AGGREGATE and COMBINE operations can be written as a single operation. Variants of GCN may consider $K$-hop graph convolutions instead of one-hop (Defferrard et al., 2016; Gama et al., 2018; Du et al., 2018).

**GraphSAGE.** The layerwise propagation rule of GraphSAGE is given by (Hamilton et al., 2017)

$$\begin{aligned} \mathbf{h}_{N(v)}^{(\ell+1)} &= \text{AGGREGATE}_\ell\left(\{\mathbf{h}_u^{(\ell)}, u \in N(v)\}\right) \\ \mathbf{h}_v^{(\ell+1)} &= \sigma\left(\mathbf{W}^{(\ell+1)} \cdot \text{CONCAT}\left(\mathbf{h}_v^{(\ell)}, \mathbf{h}_{N(v)}^{(\ell+1)}\right)\right) \qquad \text{and} \qquad \mathbf{h}_v^{(0)} = \mathbf{x}_v, \end{aligned} \tag{9}$$

where $\mathbf{W}^{(\ell+1)} \in \mathbb{R}^{F_{\ell+1} \times 2F_\ell}$ are learnable weights and $\sigma$ is a pointwise nonlinearity. Typical AGGREGATE operations are the mean, the sum, and the max.

## C  CLASSICAL SAMPLING-BASED GNN TRAINING ALGORITHM

---
**Alg. 3:** Classical Training

---
**while** $|\nabla \mathcal{L}| > \epsilon$ **do**

    $\mathbf{Z} = \texttt{SamplingBasedGNN}(G)$

    $\nabla_{\mathbf{W}_t} \tilde{\mathcal{L}}_{\nu_g}(\mathbf{Z}) = \frac{1}{|V_B|} \sum_{v \in V_B} \frac{1}{\nu_g(v)} \nabla \mathcal{L}(\mathbf{z}_v)$

    $\mathbf{W}_{t+1} = \mathbf{W}_t - \eta \nabla_{\mathbf{W}_t} \nabla \tilde{\mathcal{L}}_{\nu_g}$

    $t \leftarrow t + 1$

---

## D  EXAMPLES OF ALMOST LOCAL FUNCTIONS

**Proposition D.1.** *For a sequence of convergent graphs $\{G_n\}_{n \in \mathbb{N}}$, the following holds.*

*1. Negative sampling: Let $\sigma$ be the Sigmoid function and $x \cdot y$ be the dot-product of two vectors $x$ and $y$, and let $g_\ell : \mathcal{G}_* : \mathbb{R}^K$ be a bounded and continuous function on rooted graphs with a finite radius $\ell \geq 0$. Then the functions $f(G_n, v, u) = \sigma\big(g_\ell(G, v) \cdot g_\ell(G, u)\big)$ and $\tilde{f}(G_n, v) = \mathbb{E}_{u \sim D_n}[f(G_n, v, u)]$ are almost local. Here $D_n$ is a distribution on nodes of $G_n$ such that its density, i.e., $n\mathbb{P}_{v \sim D_n}$, is local.*

*2. Normalized adjacency matrix: When the degree sequence is uniformly integrable[3], then re-weighting any local function with respect to the normalized adjacency matrix is almost local. Formally, if $g$ is a bounded and continuous function, then the function $f(G_n, v) = g(G_n, v) \frac{deg(v)}{\frac{1}{|V(G)|} \sum_u deg(u)}$ is almost local, and the limit can be written as,*

$$\mathbb{E}_{v \sim \mathcal{P}_n}[f(G_n, v) | G_n] \xrightarrow{\mathbb{P}} \mathbb{E}_{(G,o) \sim \mu}\Big[g(G, o) \frac{deg(o)}{\bar{d}}\Big].$$

*Proof of Proposition D.1.* **Part 1 (Negative sampling).** We start with the proof for negative sampling. The main idea is that, the value of $g_\ell(G, v)$ only depends on $\ell$ neighborhood of the node $v$. On the other hand, given the assumption on $D_n$ two random nodes drawn independently from $D_n$ are with high probability more than $2\ell$ far apart. As a result, their corresponding node embedding $z_v = g_\ell(G, v)$ and $z_u = g_\ell(G, u)$ is with high probability independent of each other. So, we can use the fact that if two independent random variables converge in probability, then their product also converges.

Next, we formalize this intuition using the second-moment method. The convergence of the first moment is trivial by the

$$\mathbb{E}_{G_n}\big[\mathbb{E}_{u,v \sim \mathcal{P}_n}[f(G_n, u, v) | G_n]\big] = \mathbb{E}_{G_n}\big[\mathbb{E}_{u,v \sim \mathcal{P}_n}[\sigma(g_v \cdot g_u) | G_n]\big]$$

I want to prove it converges to

$$\mathbb{E}_{(G^{(1)}, o_1), (G^{(2)}, o_2) \sim \mu}\big[\sigma(g(g(G^{(1)}, o_1), g(G^{(2)}, o_2)))\big]$$

**Part 2 (Normalized adjacency matrix).** The proof follows the classic proofs on the locality of functions such as centrality coefficients as in (van der Hofstad, 2021, Chapter 2.4). First, note that we can rewrite:

$$\mathbb{E}_{v \sim \mathcal{P}_n}[f(G_n, v) | G_n] = \frac{\mathbb{E}_{v \sim \mathcal{P}_n}[deg(v) g(G_n, v) | G_n]}{\mathbb{E}_{v \sim \mathcal{P}_n}[deg(v) | G_n]}. \tag{10}$$

The local convergence in probability (6) applies to only bounded and continuous functions. However, the graph might have unbounded degrees that could make the enumerator or the denominator of (10) unbounded. We will control its effect using the uniform integrability assumption on the degree sequence.

First, we will focus on the enumerator. In particular, for any fixed integer $\Delta > 0$, we can split

$$\mathbb{E}_{v \sim \mathcal{P}_n}[deg(v) g(G_n, v) | G_n] = \mathbb{E}_{v \sim \mathcal{P}_n}[deg(v) g(G_n, v) 1(deg(v) \leq \Delta) | G_n]$$
$$+ \mathbb{E}_{v \sim \mathcal{P}_n}[g(G_n, v) deg(v) 1(deg(v) > \Delta) | G_n].$$

---

[3]A random variable $X$ is said to be uniformly integrable if the probability that it exceeds a certain threshold value approaches zero as the threshold increases towards infinity, i.e., for any $\epsilon > 0$ there exists $K_\epsilon$ such that $\mathbb{P}(X > K_\epsilon) < \epsilon$.

The first term, $deg(v)g(G_n,v)1(deg(v) \leq \Delta)$, is already a bounded continuous function and we can apply (6), to get

$$\mathbb{E}_{v \sim \mathcal{P}_n}[deg(v)g(G_n,v)1(deg(v) \leq \Delta)|G_n] \xrightarrow{\mathbb{P}} \mathbb{E}_{(G,o) \sim \mu}[deg(o)g(G,o)1(deg(o) \leq \Delta)].$$

For the second term, we bound it using uniform integrability. Since $g$ is a bounded function, there exists $M$ as an upper bound for it. The uniform integrability implies that for any $\epsilon > 0$ there exists a large enough $N_\epsilon$, such that for all $n > N_\epsilon$,

$$\mathbb{E}_{v \sim \mathcal{P}_n}[deg(v)g(G_n,v)] \leq \mathbb{E}_{v \sim \mathcal{P}_n}[deg(v)M] \leq \epsilon^2. \tag{11}$$

Then using Markov inequality,

$$\mathbb{P}\Big(\mathbb{E}_{v \sim \mathcal{P}_n}[deg(v)g(G_n,v)] \geq \epsilon|G_n\Big) \leq \frac{1}{\epsilon}\mathbb{E}_{v \sim \mathcal{P}_n}[deg(v)M] \leq \epsilon. \tag{12}$$

Therefore,

$$\mathbb{E}_{v \sim \mathcal{P}_n}[deg(v)g(G_n,v)|G_n] \xrightarrow{\mathbb{P}} \mathbb{E}_{(G,o) \sim \mu}[deg(o)g(G,o)].$$

Similarly, we can get

$$\mathbb{E}_{v \sim \mathcal{P}_n}[deg(v)|G_n] \xrightarrow{\mathbb{P}} \mathbb{E}_{(G,o) \sim \mu}[deg(o)],$$

which would prove the second part of the proposition. $\qquad\square$

# E   PROOF OF THEOREM 5.4

We begin the proof by relating the gradient of the loss function on finite samples to the gradient of the loss function in the limit. To formalize the proof, let $\mathcal{L}_{\nu_g}(W_t, G)$ be the loss of applying $W_t$ (coefficients at iteration $t$ of the algorithm) to graph $G$ with node sampling $\nu_g$ [4].

In this section, we present the proof of Theorem 5.4, which provides a theoretical guarantee for the convergence of the sampling-based GNN training algorithm presented in Algorithm 2.

To start the proof, we first establish a connection between the gradient of the loss function on finite samples and the gradient of the loss function in the limit. Specifically, we use the notation $\mathcal{L}_{\nu_g}(W_t, G)$ to denote the loss of applying the coefficients $W_t$ at iteration $t$ to a graph $G$ with node sampling $\nu_g$ [5].

Our proof proceeds in two main steps. First, we show that in each iteration of the algorithm, the gradient of the loss on the sampled graphs is close to the expected gradient of the loss in the limit infinite graph (Lemma E.1). Second, we show that the loss on finite graphs decreases at each step with a high probability for the correct choice of learning rates (Lemma E.3).

**Lemma E.1.** *Given the assumptions of Theorem 5.4, for any $t > 0$, then the loss in iteration $t$ of Algorithm 2 on a finite graph $G_n$ with appropriately sized mini-batches is with high probability close to the expected loss on the limit. In particular, for any $\epsilon > 0$ there exists $N_\epsilon$ and $\mathcal{Y}_\epsilon$, such that for $n > N_\epsilon$, when either the mini-batch is large enough (i.e., $V_B \geq \mathcal{Y}_\epsilon$) but at the same time not too large (i.e., $V_B = o(\sqrt{|V(G_n)|})$) or includes all nodes (i.e., $|V_B| = |V(G_n)|$),*

$$\mathbb{P}\Big(|\nabla\tilde{\mathcal{L}}_{\nu_g}(W_t, G_n) - \mathbb{E}_\mu[\nabla\mathcal{L}(W_t, G)]| \geq \epsilon\Big) \leq \epsilon,$$

*where the probability is over the possible randomness of $G_n$, the randomness of SGD, and the computational graph sampler $\nu_C$.*

*Remark* E.2. The Lemma requires an upper bound on the minibatch size to ensure that the embeddings and their loss are independent. This is necessary to control for the possibility of the local neighborhood of sampled nodes not being disjoint.

---

[4] For simplicity, we don't show the input feature in this representation. But all results apply to convergent graphs with input features.

[5] Note that for simplicity, we do not explicitly show the input feature in this notation, but all results apply to convergent graphs with input features.

*Proof.* First, consider the case that the minibatch $V_B$ includes all nodes, i.e., we use gradient descent instead of SGD to compute the gradient of loss. Then, the loss on the finite graph converges to the limit by the assumption that the loss is almost local. In particular, for any $\epsilon' > 0$ there exist $N_{\epsilon'}$ such that for $n > N_{\epsilon'}$,

$$\mathbb{P}\Big(|\nabla\tilde{\mathcal{L}}(W_t, G_n) - \mathbb{E}_\mu[\nabla\mathcal{L}(W_t, G)]| \geq \epsilon'\Big) \leq \epsilon', \tag{13}$$

where the probability is over the possible randomness of $G_n$, and the weights $W_t$ (which in turn depends on the randomness of the computational graph sampler $\nu_C$). In fact, as the size of the graphs in the sequence $G_{n\,n\in\mathbb{N}}$ grows, we have convergence in probability,

$$\nabla\tilde{\mathcal{L}}(W_t, G_n) \xrightarrow{\mathbb{P}} \mathbb{E}_\mu[\nabla\mathcal{L}(W_t, G)].$$

Note that there is no expectation on the left-hand side of neither Lemma E.1 nor (13) nor the above expression, as the loss (and its gradient) are invariant to the choice of root in $G_n$.

Next, we consider the case where $V_B$ is an i.i.d subset of nodes. The idea is to use of the fact that the loss is almost local, and hence only depends on a bounded neighborhood of a few nodes. So, if we draw a small enough minibatch $|V_B| = o(\sqrt{|V(G_n)|})$ then the loss of different nodes should be independent of each other, and we can use Hoeffding bound to control the error. We formalize this idea next.

Let $v_1, \ldots, v_{V_B}$ be the sampled nodes in the minibatch. First, note that since the loss is almost local, there exists $K_\epsilon$ such that there exists a function on $K_\epsilon$ neighborhood of nodes which is a $(1 - \epsilon/4)$ approximation of the loss. We continue by proving that the local neighborhoods of the sampled nodes in the minibatch are disjoint with high probability. For this purpose, let $I_{V_B} = \{\forall i, j \in [V_B], \quad B_K(G_n, v_i) \cap B_K(G_n, v_i) = \emptyset\}$ be the event that $K$-neighborhoods are disjoint. We want to prove that there exists $N_\epsilon$ such that for $|V(G_n)| \geq N_\epsilon$ and any $|V_B| = o(\sqrt{|V(G_n)|})$,

$$\mathbb{P}(I_{V_B} \text{ does not happen.}) \leq \frac{\epsilon}{4}. \tag{14}$$

Define $V_{K,\Delta}$ as the set of nodes such that the maximum degree in their $K$ neighborhood is at most $\Delta$. Since the sequence $\{G_n\}_{n\in\mathbb{N}}$ converges in the local sense, for all $\epsilon > 0$, there exists $\Delta < \infty$ and $N'_\epsilon < \infty$ such that for $n \geq N'_\epsilon$, with probability $1 - \frac{\epsilon}{4}$ we have $\frac{|V_{k_\epsilon,\Delta}|}{n} \geq 1 - \frac{\epsilon}{4}$. Let $E_\epsilon$ be the event that $\frac{|V_{k_\epsilon,\Delta}|}{n} \geq 1 - \frac{\epsilon}{4}$. Then by a union bound

$$\mathbb{P}(I_{V_B} \text{ does not happen}) \leq \mathbb{P}(I_{V_B} \text{ does not happen} \mid E_\epsilon) + \mathbb{P}(E_\epsilon) \leq |V_B|^2 \frac{\Delta^K}{|V(G_n)|} + \frac{\epsilon}{8}.$$

Since $\Delta$ and $K$ are independent from $|V(G_n)|$, by increasing $N'_\epsilon$, if necessary, we can assume that $|V_B|\Delta^K \leq \frac{\epsilon}{8}|V(G_n)|$ and hence proving (14).

Now, conditioned on the event $I_{V_B}$, we can apply Hoefdding bound. Let $X_i = \mathcal{L}(G_n, v_i)$, where $v_i \sim \mathcal{P}_n$ is drawn uniformly at random. Then,

$$\mathbb{P}(|\nabla\tilde{\mathcal{L}_{V_B}} - \nabla\tilde{\mathcal{L}_{V_B}}| \geq \frac{\epsilon}{2}) \leq \mathbb{P}(|\nabla\tilde{\mathcal{L}_{V_B}} - \nabla\tilde{\mathcal{L}_{V_B}}| \geq \frac{\epsilon}{2} \mid I_{V_B}) + \mathbb{P}(I_{V_B} \text{ does not happen})$$

$$\leq exp(-\frac{|V_B|\epsilon}{C^2}) + \frac{\epsilon}{4}.$$

So, if we choose $|V_B|$ between $o(\sqrt{|V(G_n)|})$ and $\Omega(\log(1/\epsilon))$, we will get

$$\mathbb{P}(|\nabla\tilde{\mathcal{L}_{V_B}} - \nabla\tilde{\mathcal{L}_{V_B}}| \geq \frac{\epsilon}{2}) \leq \frac{\epsilon}{2}.$$

This together with (13) gives the result. $\qquad\square$

In the following lemma, we demonstrate that the loss in the limit decays in each iteration of the algorithm by applying the above lemma. Notably, we only rely on the Lipschitz property of the loss function in our proof and do not make any other use of the locality of the loss function, as long as Lemma E.1 holds.

**Lemma E.3.** *Given the assumption of Theorem 5.4, fix some $\epsilon > 0$. Also assume the learning rate $\eta$ is smaller than $1/C$, where $C$ is the loss function's Lipschitz constant. Then there exists some $N_\epsilon$ such that if Algorithm 2 is trained on graphs of size larger than $N_\epsilon$, then the loss on the limiting graph decreases in each iteration,*

$$\mathbb{P}\Big(\mathbb{E}_\mu[\mathcal{L}(W_{t+1}, G)] \leq \mathbb{E}_\mu[\mathcal{L}(W_t, G)] - \frac{\eta}{4}||\mathbb{E}_\mu[\nabla\mathcal{L}(W_t, G)]||^2\Big) \geq 1 - \epsilon,$$

*where the probability is over the possible randomness of $G_n$, the randomness of SGD, and the computational graph sampler $\nu_C$.*

*Proof.* We use a helper function to smooth out the difference between the two steps of the loss function. Let (in the following the expectation is over the limit $G$),

$$g(\epsilon) = \mathbb{E}_\mu[\mathcal{L}(W_t - \epsilon\eta\nabla\mathcal{L}(W_t, G_n), G))].$$

Then $g(1) = \mathbb{E}[\mathcal{L}(W_{t+1}, G)]$ and $g(0) = \mathbb{E}[\mathcal{L}(W_t, G)]$. This definition of a helper function has been classically used in the literature to prove convergence of the loss (Cerviño et al., 2023; Bertsekas & Tsitsiklis, 2000).

By differentiating the helper function, $\frac{\partial g}{\partial \epsilon} = -\eta\nabla\mathcal{L}(W_t, G_n)\mathbb{E}_\mu[\nabla\mathcal{L}(W_t - \epsilon\eta\mathcal{L}(W_t, G_n), G))]$. So, we can write,

$$g(1) - g(0) = \int_0^1 \frac{\partial g}{\partial \epsilon} d\epsilon.$$

$$= -\eta\nabla\mathcal{L}(W_t, G_n)\int_0^1 \mathbb{E}_\mu[\nabla\mathcal{L}(W_t - \epsilon\eta\mathcal{L}(W_t, G_n), G))]d\epsilon$$

Then we add and subtract $\mathbb{E}_\mu[\nabla\mathcal{L}(W_t, G)]$ to get

$$\mathbb{E}_\mu[\mathcal{L}(W_{t+1}, G)] - \mathbb{E}_\mu[\mathcal{L}(W_t, G)]$$

$$= -\eta\nabla\mathcal{L}(W_t, G_n)\mathbb{E}[\nabla\mathcal{L}(W_t, G)] + \eta\nabla\mathcal{L}(W_t, G_n)\Big(\int_0^1 \nabla\mathcal{L}(W_t - \epsilon\eta\nabla\mathcal{L}(W_t, G_n), G)$$

$$- \mathbb{E}_\mu[\nabla\mathcal{L}(W_t, G)]d\epsilon\Big)$$

Now since $\nabla\mathcal{L}$ is Lipschitz (with constant $C$),

$$\mathbb{E}_\mu[\mathcal{L}(W_{t+1}, G)] - \mathbb{E}_\mu[\mathcal{L}(W_t, G)]$$

$$\leq -\eta\nabla\mathcal{L}(W_t, G_n)\mathbb{E}[\nabla\mathcal{L}(W_t, G)] + \eta|\nabla\mathcal{L}(W_t, G_n)|\Big(\int_0^1 \epsilon\eta C||\nabla\mathcal{L}(W_t, G_n)||d\epsilon\Big)$$

$$= -\eta\nabla\mathcal{L}(W_t, G_n)\mathbb{E}[\nabla\mathcal{L}(W_t, G)] + \frac{\eta^2 C}{2}||\nabla\mathcal{L}(W_t, G_n)||^2$$

$$= \frac{\eta^2 C - \eta}{2}||\nabla\mathcal{L}(W_t, G_n)||^2 - \frac{\eta}{2}\Big(||\mathbb{E}_\mu[\nabla\mathcal{L}(W_t, G)]||^2 - ||\nabla\mathcal{L}(W_t, G_n) - \mathbb{E}_\mu[\nabla\mathcal{L}(W_t, G)]||^2\Big).$$

To finish the proof, we can bound the first term by choosing the learning rate $\eta$ smaller than $1/C$, and the second term by using Lemma E.3. $\qquad\square$

Now, we are ready to prove the theorem. For proof, we analyze the stopping time $t^*$, which is the first iteration at which the expected gradient of the loss with respect to the coefficients falls below a threshold $\epsilon$. Then we use Lemma E.3 to bound this stopping time.

*Proof of Theorem 5.4.* Given $\epsilon > 0$ define the stopping time

$$t^* = \inf_t\{\mathbb{E}_{\mu, W_t}(\nabla\mathcal{L}(W_t, G)) \leq \epsilon\},$$

where the expectation is both over the randomness of the limit $G$ and the coefficients $W_t$. Note that the randomness of $W_t$ is due to the randomness of sampling the graph $G_n$. We write loss as the sum of differences of loss in each iteration,

$$\mathbb{E}_{\mu, W_t}\big(\mathcal{L}(W_0, G) - \mathcal{L}(W_{t^*}, G)\big) = \mathbb{E}_{\mu, W_t}\Big(\sum_{t=0}^{t^*-1} \mathcal{L}(W_t, G) - \mathcal{L}(W_{t+1}, G)\Big).$$

We can take the expected value with respect to the randomness of $t^*$,

$$\mathbb{E}_{t^*}\mathbb{E}_{\mu,W_t}\big(\mathcal{L}(W_0,G) - \mathcal{L}(W_{t^*},G)\big) = \sum_{t^*=0}^{\infty}\mathbb{E}_{\mu,W_t}\big(\sum_{t=0}^{t^*-1}\mathcal{L}(W_t,G) - \mathcal{L}(W_{t+1},G)\big)\mathbb{P}(t^*).$$

By applying Lemma E.3 for $t < t^*$,

$$\mathbb{P}\Big(\mathbb{E}_\mu\big(\mathcal{L}(W_t,G) - \mathcal{L}(W_{t+1},G)\big) \geq \frac{\eta}{4}\epsilon^2\Big) \geq 1 - \epsilon.$$

By applying this to the previous inequalities:

$$\mathbb{E}_{t^*}\mathbb{E}_{\mu,W_t}\big(\mathcal{L}(W_0,G) - \mathcal{L}(W_{t^*},G)\big) \geq \frac{\eta\epsilon^2(1-\epsilon)}{4}\sum_{t^*=0}^{\infty}t^*\mathbb{P}(t^*) = \frac{\eta\epsilon^2(1-\epsilon)}{4}\mathbb{E}[t^*].$$

Since loss is non-negative

$$\frac{4}{\eta\epsilon^2(1-\epsilon)}\mathbb{E}_{t^*}\mathbb{E}_\mu\big(\mathcal{L}(W_0,G)\big) \geq \mathbb{E}[t^*].$$

$\square$

# F  APPLICATIONS

Our main result applies to various GNN architectures, including GCN, GraphSAGE, and GIN with mean readout; and various sampling mechanisms, such as neighborhood sampling (GraphSAGE), FastGCN, and shaDoW-GNN. In each of the following sections, we will discuss how each model fits into our unified framework (Algorithm 2), and explain why they satisfy the assumptions of the main theorem.

**GCN.** The GCN architecture (Kipf & Welling, 2017), as described in (8), applies the convolutional layer on the entire computational graph of a node. So, for a $L$-layer GCN and a given node $v$, the computational graph sampler $\mu_C(v)$ returns the entire $L$-neighborhood of $v$. In addition, it is common to use SGD to compute the gradient in GCNs. Both the gradient and computational graph samplers satisfy the locality assumption 5.2, and hence, our main theorem applies.

**GIN.** In the Graph Isomorphism Network (GIN) (Xu et al., 2019), the AGGREGATE and COMBINE operations consist of a multi-layer perceptron applied to the sum of each node's embeddings with their neighbors' embeddings. In its standard form, GIN thus takes in the entire computational graph of a node. Since the AGGREGATE and COMBINE operations are local, in a node-level task GIN satisfies all of our assumptions, hence our results hold. In a graph-level task, our results hold provided that the readout is a mean aggregation, which is permutation invariant and, unlike the sum, does not increase with the graph size.

**GraphSAGE with neighborhood sampling.** GraphSAGE (Hamilton et al., 2017) is a popular GNN architecture that generates node embeddings by concatenating information from each node's local neighborhood. They also propose to train GraphSAGE with a computational graph sampling technique called neighborhood sampling, where they sample a fixed number of neighbors for each node. The computational graph sampler $\nu_C$ assigns probabilities proportional to $1/\binom{deg(v)}{K_\ell}$ to all sets of size $K_\ell$ from the neighbors of node $v$ at layer $\ell$, where $K_\ell$ is the number of nodes to sample at layer $\ell$ and $deg(v)$ is the degree of node $v$. They also use SGD for gradient sampler $\nu_g$.

Another novelty of the GraphSAGE approach was to suggest unsupervised learning based on computing loss with negative sampling, which is almost local per Proposition D.1. Therefore, our result applies to both their semi-supervised and unsupervised training.

**FastGCN.** Fast GCN (Chen et al., 2018) relies on layerwise sampling to address scalability issues GNNs. The computational graph sampler in FastGCN subsamples nodes from each layer based on the normalized adjacency matrix, as expressed by the equation

$$q(u;v) = \frac{\|\mathbf{A}'(u,v)\|^2}{\|\sum_{u'\in N(v)}\mathbf{A}'(u',v)\|^2},$$

which by Proposition D.1 is almost local. So, for a node $v$ already sampled in the computational graph, $\nu_C$ samples a fixed number of nodes $k_\ell$ in the next layer w.r.t $q(u; v)$. This architecture also uses SGD as the gradient sampler.

**shaDoW-GNN.** shaDoW-GNN (Zeng et al., 2021) is a method that tackles the scalability issue of GNNs by subsampling a subgraph for each node in a minibatch. Specifically, for a $L'$-layer GNN, shaDoW-GNN selects a subgraph with nodes that are at most $L$ hops away from each node in the minibatch, where $L \leq L'$. They decouple nodes that appear in more than one sampled subgraph by keeping two copies of them. Their framework allows either to keep the whole $L$-neighborhood or to sample from it, similar to GraphSAGE or FastGCN. So, the computational graph sampler $\nu_C$ is similar to one of the previous frameworks, with the only difference being that it creates new copies in memory for each node sampled multiple times.

**GNNAutoScale.** A recent addition to the family of GNNs is GNNAutoScale(Fey et al., 2021), which integrates well with our framework. It leverages historical embeddings of out-of-sample neighbors during training, merging minibatch sampling and historical embeddings. This method aligns well with our framework for the following reason: Each iteration of embedding calculation is convergent, as shown by Lemma **??**. So, one can use historical embeddings as 'features' for subsequent iterations during information aggregation, so it satisfies the Assumption 5.2. Therefore, we can view GNNAutoScale's approach through the lens of our framework.

**GraphFM.** This framework by (Yu et al., 2022), although bearing similarities with GNNAutoScale, distinguishes itself by including historical embeddings of nodes within the one-hop boundary of selected minibatches. This novel incremental update strategy remains consistent with the principles of our framework, indicating a promising compatibility.

**LMC.** Another model, LMC(Shi et al., 2022), while echoing GNNAuto-scale in many respects, shows unique differences in aggregation during its forward/backward propagation. Its localized aggregation strategies remain consistent with our proposed framework, satisfying with the assumptions of our result.

**IBMB.** This method by (Gasteiger et al., 2022) employs a distinctive approach by computing influence scores for nodes and subsequently optimizing the selection of influential nodes for computation. Despite its broad methodology of calculating node influence, the implementation specifics, especially using pagerank computation and localized node selection, align well with our framework (given that pagerank is a local function as shown in (Garavaglia et al., 2020)).

## G  EXPERIMENT DETAILS

**Citation networks.** Cora, CiteSeer, PubMed and ogbn-mag are citation networks commonly used to benchmark GNNs. Their nodes represent papers, and their edges are citations between papers (in either direction). Each paper is associated with a bag-of-words vector, and the task is to use both the citation graph and the node information to classify the papers into $C$ classes. Relevant dataset statistics are presented in Table 1.

**Training details.** In all experiments, we use PyTorch Geometric (Fey & Lenssen, 2019) and consider a GraphSAGE (Hamilton et al., 2017) and a GCN (Kipf & Welling, 2017) architectures with 2 layers and embedding dimensions 64 and 32 respectively (64 and 64 for ogbn-mag). In the first and the second layers, the nonlinearity is a ReLU, and in the readout layer, a softmax. We minimize the negative log-likelihood (NLL) loss and report the classification accuracy. We consider the train-validation-test splits from the full Planetoid distribution from PyTorch Geometric for PubMed, Cora, and CiteSeer; and from Open Graph Benchmark for ogbn-mag. To train the models, we use ADAM with the standard forgetting factors (Kingma & Ba, 2015).

## H  ADDITIONAL EXPERIMENTS ON CORA AND CITESEER

In this section, we repeat the experiments of Section 6 for the Cora and CiteSeer datasets, whose corresponding dataset statistics can be found in Table 1. The results for node sampling, computational graph sampling, and no sampling are described below.

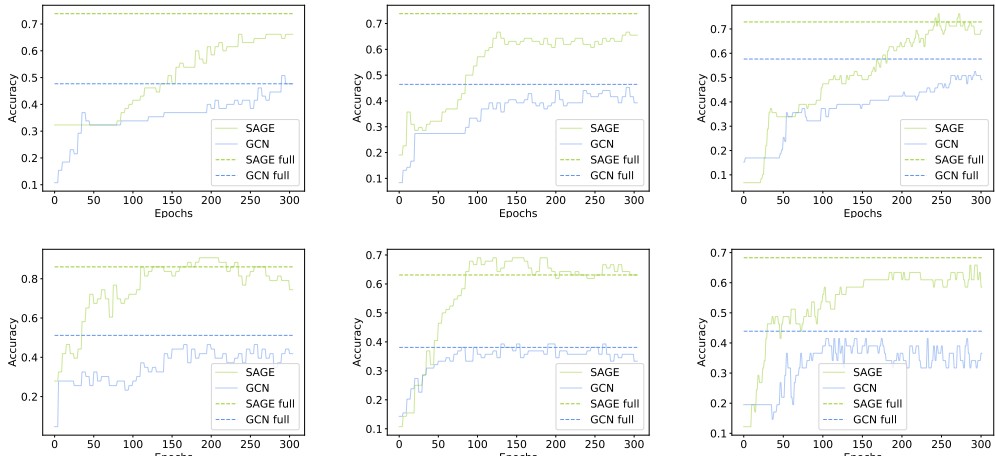

Figure 4: Node sampling results for Cora (top) and CiteSeer (bottom) with batch size 32. We consider three scenarios in terms of the small graph size $n$ and the graph sampling interval $\gamma$: $n = 300$, $\gamma = 5$ epochs (left); $n = 500$, $\gamma = 5$ epochs (center); $n = 300$, $\gamma = 2$ epochs (right). Note that these graphs have size equal to approximately 10-16% of the original graph size.

Table 1: Dataset statistics.

|          | Nodes ($N$) | Edges      | Features | Classes ($C$) |
|----------|-------------|------------|----------|---------------|
| Cora     | 2708        | 10556      | 1433     | 7             |
| CiteSeer | 3327        | 9104       | 3703     | 6             |
| PubMed   | 19717       | 88648      | 500      | 3             |
| ogbn-mag | 1,939,743   | 21,111,007 | 128      | 349           |

**Node sampling.** In Figure 4, we consider three scenarios in terms of the small graph size $n$ and the graph sampling interval $\gamma$: $n = 300$, $\gamma = 5$ epochs (left); $n = 500$, $\gamma = 5$ epochs (center); $n = 300$, $\gamma = 3$ epochs (right). Note that these graphs have size equal to approximately $10 - 16\%$ of the original graph size. The GNNs are trained for 300 epochs with a learning rate $1e-4$ and batch size 32. In Figure 4, we observe that resampling graphs every $\gamma = 5$ epochs, at either $n = 300$ (left) or $n = 500$ (center), is not enough to train both models (but especially GraphSAGE) to correctly classify nodes on the full Cora graph under 300 epochs. However, decreasing the resampling interval to $\gamma = 2$ (right) helps. In the case of CiteSeer, $n = 300$ and $\gamma = 5$ (left) are enough to match the accuracy of the model trained on the full graph, but the models learn faster when $n = 500$ (center). Increasing the sampling rate (right) increases the variability in accuracy and worsens performance.

**Computational graph sampling.** In this experiment, we fix the neighborhood sizes for computational graph sampling in both layers at $K_1 = K_2 = 32$, and also consider node sampling with batch size 32. The combinations of graph size $n$ and sampling interval $\gamma$, and the number of epochs and the learning rate are the same as in the node sampling experiment. The results are reported in Figure 5. For Cora, we observe the best results for $n = 500$ and $\gamma = 5$ (center), and for CiteSeer, for $n = 300$ and $\gamma = 5$ (left). On both datasets, increasing the sampling rate (right) increases the variability in performance, which is undesirable. In the case of CiteSeer, increasing the graph size (center) leads to some overfitting for GraphSAGE.

# I EXPERIMENTS WITHOUT NODE AND COMPUTATIONAL GRAPH SAMPLING

Here, we consider GNNs without any form of sampling other than the $n$-node random graph sequences sampled from the target graph. The combinations of graph size $n$ and sampling interval $\gamma$ are the same as in the two previous experiments, and the GNNs are trained for 300 epochs with a learning rate $1e-3$. The results are reported in Figure 6 for PubMed and in Figure 7 for Cora and CiteSeer. The GraphSAGE models trained on the random graph sequences generally achieve better performance on

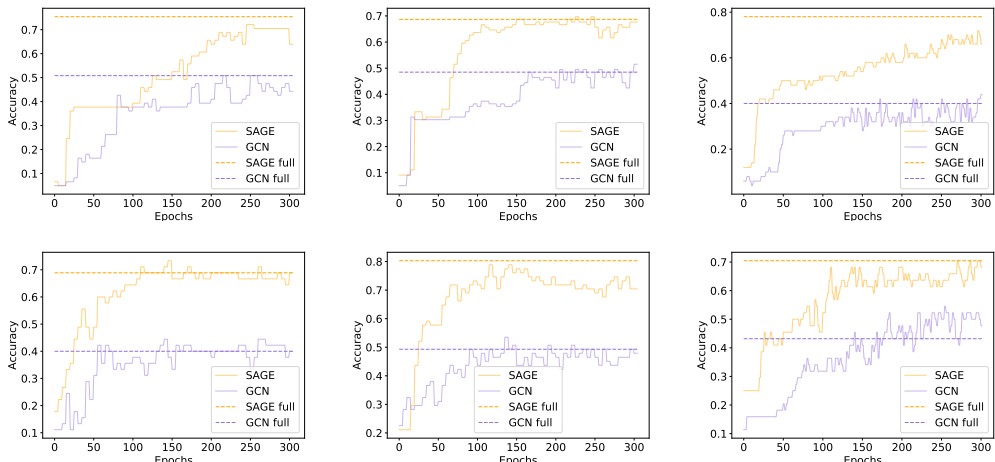

Figure 5: Computational graph sampling results for Cora (top) and CiteSeer (bottom) with $K_1 = K_2 = 32$ and batch size 32. We consider three scenarios in terms of the small graph size $n$ and the graph sampling interval $\gamma$: $n = 300$, $\gamma = 5$ epochs (left); $n = 500$, $\gamma = 5$ epochs (center); $n = 300$, $\gamma = 2$ epochs (right). Note that these graphs have size equal to approximately 10-16% of the original graph size.

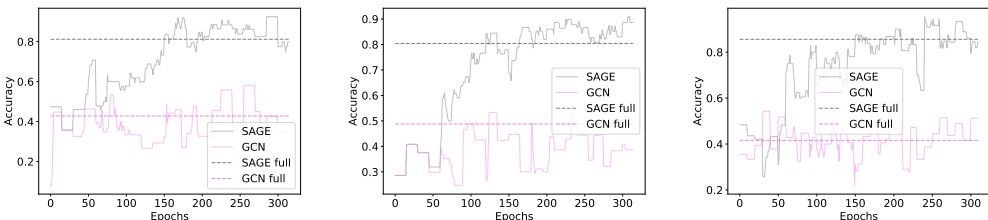

Figure 6: Results for PubMed without any sampling other than the graph sequence sampling. We consider three scenarios in terms of the small graph size $n$ and the graph sampling interval $\gamma$: $n = 1500$, $\gamma = 15$ epochs (left); $n = 2000$, $\gamma = 15$ epochs (center); $n = 1500$, $\gamma = 10$ epochs (right). Note that these graphs have size equal to approximately 10% of the original graph size.

the random graph sequences than on the target graph, with slight accuracy improvement when $n$ is increased and higher variability in accuracy when $\gamma$ is decreased. The GCN performance is subpar in both cases and for all combinations of $n$ and $\gamma$. We observe more variability in accuracy than in Figures 1 and 2, which is expected since in the absence of node sampling, the gradient updates are calculated at all nodes; and in the absence of computational graph sampling, the effective graph neighborhoods are less regular. It is also interesting to note that gradient and computational graph sampling provide good inductive bias in this experiment, as the test accuracy achieved by the GNNs with node and computational graph sampling, in Figures 1,4 and 2,5, are higher than those achieved by the GNNs without sampling in Figures 6–7.

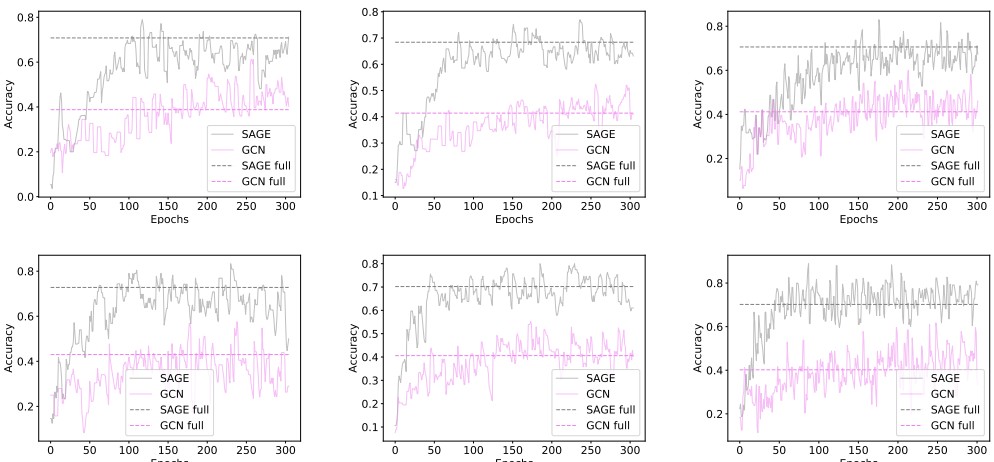

Figure 7: Results for Cora (top) and CiteSeer (bottom) without any sampling other than the graph sequence sampling. We consider three scenarios in terms of the small graph size $n$ and the graph sampling interval $\gamma$: $n = 300, \gamma = 5$ epochs (left); $n = 500, \gamma = 5$ epochs (center); $n = 300, \gamma = 2$ epochs (right). Note that these graphs have size equal to approximately 10-16% of the original graph size.

