# OpenReview forum: "A Local Graph Limits Perspective on Sampling-Based GNNs"
_ICLR.cc/2024/Conference — Submitted to ICLR 2024_

### Official Review · Reviewer_HSq8 · 2023-10-29

**Soundness:** 3 good
**Presentation:** 3 good
**Contribution:** 2 fair
**Rating:** 5
**Confidence:** 4

**Summary:**

This paper propose to accelerate the training of Graph Neural Networks (GNNs) by training GNNs on small subgraphs insead of the whole large graph. Moreover, the authors show that the parameters of GNNs trained on small subgraphs are within an $\epsilon$-neighborhood of that trained on the whole graph.

**Strengths:**

1. This paper is easy to follow.
2. The authors provide the theoretical analysis of the proposed method.

**Weaknesses:**

1. Please explain why Assumption 5.3 holds in practice.
2. The proposed method aims to train GNNs on large graphs efficiently, while the authors does not evaluate the runtime and memory in experiments.
3. The authors claim that no formal models explain why they can reduce the computation and memory requirements of GNNs without signifcantly compromising their performance. However, in my opinion, VR-GCN [1] and LMC [2] explain the phenomenon of node-wise sampling and subgraph-wise sampling respectively in terms of convergence.


[1] Stochastic Training of Graph Convolutional Networks with Variance Reduction. ICML 2018.

[2] LMC: Fast Training of GNNs via Subgraph Sampling with Provable Convergence. ICLR 2023.

**Questions:**

See Weaknesses.

---

### Official Review · Reviewer_pWTk · 2023-10-31

**Soundness:** 3 good
**Presentation:** 3 good
**Contribution:** 3 good
**Rating:** 6
**Confidence:** 3

**Summary:**

This paper proposes a unified framework for sampling-based GNNs, and then uses Local Graph Limits theory to show that training on small sampled subgraphs guarantee convergence to eps-neighborhood of the solution trained on the entire graphs.

**Strengths:**

- Timely contribution as graphs are getting bigger and bigger and computational resources become tighter.
- Nice theory for the convergence.
- Works for different types of tasks.

**Weaknesses:**

- Missing literature, e.g., https://arxiv.org/pdf/2006.13866.pdf and related papers.
- Experiments could be more comprehensive by looking at more datasets and more algorithms.
- The unification is not as novel as advertised. Since available platforms support sampling, implicitly the framework has been established before. See, e.g., https://arxiv.org/pdf/2207.03522.pdf.

**Questions:**

- In Fig 3: why does the sampling-based method perform better than GNN trained on entire graph?

---

### Official Review · Reviewer_o4pB · 2023-11-01

**Soundness:** 4 excellent
**Presentation:** 4 excellent
**Contribution:** 2 fair
**Rating:** 6
**Confidence:** 2

**Summary:**

In this paper, the authors propose a new framework to analyze the behavior of GNN training using sampled subgraphs of an original, large input graph, which encapsulates many of the currently-used GNN algorithms. In doing so, the authors prove that using this paradigm, the parameters of a model converge within some $\epsilon$ bound around the parameters if they were trained on the original graph. Authors also include empirical results on using their new sampling-based framework, and show that indeed it is comparable to training on the original graph.

**Strengths:**

The paper's strengths lie in both its strong theoretical and empirical sides. The paper backs up any claim with solid theory, which is then compared experimentally with cohesive experiments. In addition, the paper has strong presentation; the overall flow and quality of the paper makes it very clear to read and understand.

**Weaknesses:**

The main weakness is the lack of novelty and impact the work may have on the field as a whole. Although the ideas presented in the paper are novel, they do not seem like they would have much impact in the future of the field. For example, perhaps there could be more discussion on how efficiency in both time and space are improved, practically, using the framework presented. And if the paper is purely focused on theory, perhaps there could be better bounds of existing algorithms as an example illustrating the power of the framework provided.

**Questions:**

1. Section 3.2 introduces the general framework. Does the framework only take into consideration "node sampling" and "computational graph sampling"? Would it be possible to add other forms of sampling (known currently, or in discovered in the future) easily to this framework?
2. In figure 2, right side, could you offer an intuitive explanation as to why you think GCN struggles compared to GCN full considerably more in this experiment compared to others?

---

### Official Review · Reviewer_Jmqn · 2023-11-02

**Soundness:** 2 fair
**Presentation:** 2 fair
**Contribution:** 2 fair
**Rating:** 5
**Confidence:** 5

**Summary:**

Paper is interested in sampling subgraphs for training GNNs on large input graphs, rather than doing gradient decent steps on the whole graph at every training step. They write a general training pipeline that can be applied on most message passing GNNs (with AGGREGATE and COMBINE), including GraphSAGE, GCN, and others. They theoretically show that, under some assumptions, the parameters recovered by subgraph training are not too far away form the parameters of full-graph training.

**Strengths:**

1. General algorithm that can scale many GNNs onto large graphs
2. Theoretical insight of convergence

**Weaknesses:**

# Experiments

The paper is lacking on experiments -- there are only two datasets: pubmed (small size) and ogb-mag (medium size). It is nice to show some wins on more datasets, especially larger ones. I also suggest to remove the word "very" for "very large citation networks", on 2nd page 3rd paragraph.

In fact, the SIGN model (also designed to scale GNNs, by computing pretraining ($Z = AAX$) then training on rows $Z$ without ever looking at adjacency matrix $A$ ever again during training, i.e., the graph is only used for untrainable pre-processing then usual SGD training proceeds.

# Comparison with earlier work

The work on GTTF (Marowitz et al, ICLR 2021) is very similar: it develops a general algorithm that can be specialized for training GNNs and skip-gram models (like deepwalk) by sampling, they also prove that the parameters learned with sampling are equivalent to ones learned on full graph, under some assumptions (the actual assumptions and proofs in GTTF differ).

It would be nice if this paper brings in something new: if it is the analysis, name the paper appropriately. Otherwise, it feels like just another paper that is doing sampling (like GraphSAGE, GTTF, etc)

# Uninterpretable math

* The notation is not standard. E.g., instead of $READOUT(h_v, v\in V)$, it could be $READOUT([h_v]_{v \in V})$ -- As-is, it looks like a function accepting 2 arguments: a feature vector and a boolean (is $v$ a member of $V$).

* Further, the expression for $\mathcal{P}_n$ at end of page 5 is hard to decipher

* In section 3.2, there are 3 conflicting definition of $\mathcal{v}_g$. It says $\mathcal{v}_g : G \rightarrow R^+$ but then invokes it on same line as $v_g(G, v)$

* Having READOUT function (last line of Alg.1) is ambiguous: how are the sampled nodes any special to give a signal to the entire graph?

* Is $\mu$ a measure or an instance? In definition 4.1, it overloads it to be both

* What is $B$ above Eq 6? Is it the same as defined later in Sec 4.2?

# Minor Improvements
* In Section 2: GNNAutoScale is **not** sampling-based GNN. On the other hand, it is based on historical embeddings.
* Below Equation 2, you say "simple aggregation". Why not be explicit and define "simple" right here?

**Questions:**

I have questions listed above.

---

### Meta-Review · Area_Chair_S295 · 2023-12-02

**Metareview:**

Although the paper offers intriguing theoretical groundwork, it has encountered criticism regarding its novelty and empirical evaluation. I strongly believe that the results and insights discussed would significantly benefit our research community. However, it seems that further refinement of the current manuscript may be necessary before publishing at ICLR.

**Justification For Why Not Higher Score:**

The novelty and the empirical evaluation are limited to better understand how the proposed framework can help or have effect on GNN training scheme, which limit the interest in the ICLR audience. The reviewers concerns are not addressed for further understanding and insights.

**Justification For Why Not Lower Score:**

N/A

---

### Decision · Program_Chairs · 2024-01-16

Reject